# Comparison of GC-µECD and OA-ICOS Methods for High-Precision Measurements of Atmospheric Nitrous Oxide (N₂O) at a Korean GAW Station

**Haeyoung Lee \*** 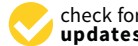**, Miyoung Ko, Sumin Kim, Wonick Seo and Young-San Park**

Innovative Meteorological Research Department, National Institute of Meteorological Sciences, Jeju 63568, Korea; miygo@korea.kr (M.K.); sulla@korea.kr (S.K.); wiseo@korea.kr (W.S.); sanpark@korea.kr (Y.-S.P.)

\* Correspondence: leehy80@korea.kr

**Abstract:** Nitrous oxide (N₂O) is a powerful greenhouse gas and is the largest remaining anthropogenic source of stratospheric ozone-depleting substances as halocarbons return towards preindustrial levels. To verify the N₂O emission inventory using inverse analysis, precise and reliable measurements are necessary. In this study, we compared the conventional gas chromatography with the microelectron capture detector method (GC-µECD, Agilent 7890A) with advanced off-axis integrated cavity output spectroscopy (OA-ICOS, Los Gatos, EP-30) for atmospheric N₂O measurements at the Jeju Gosan Suwolbong Station (JGS, 126.16° E, 33.30° N, 71.47 m a.s.l) in South Korea. The measurement uncertainties from linearity, repeatability, and reproducibility derived from the two instruments were compared. The values derived from GC-µECD were 2.4 to 8.7 times greater than that of OA-ICOS in all factors at the station. Since these factors affect the measurement quality, the calibration strategy should be well-established to reduce the measurement uncertainty. These uncertainties resulted in biases from the measurement of atmospheric N₂O. The parallel inter-comparison experiment was implemented at JGS for 22 months, and the difference in atmospheric N₂O was 0.17 ± 0.9 ppb between the two instruments. The significant differences were observed in the nonlinear range of the GC-µECD. Finally, these differences resulted in the over/underestimation of N₂O characteristics locally and seasonally. Overall, OA-ICOS has a more robust performance with a lower measurement uncertainty than GC-µECD. Based on this study, we also suggest a calibration strategy for both instruments to achieve precise N₂O measurements.

**Keywords:** GC-µECD; OA-ICOS; N₂O; GHGs; GHGs measurement

## 1. Introduction

Nitrous oxide (N₂O) is a third powerful greenhouse gas with 0.202 W·m$^{-2}$ radiative forcing (in 2019) expressed by global abundance changes relative to 1750, the preindustrial level [1] (The global warming potential of N₂O is 300 times more than that of carbon dioxide (CO₂) over a 100 year time horizon due to its long lifetime [2,3]. N₂O is also involved in stratospheric ozone depletion [4], and its emissions weighted by ozone depletion potential currently exceed those of all other substances [5].

The major sources of N₂O are microbial denitrification and nitrification, which lead to N₂O production in soils [6]; ocean waters [7,8]; and in streams, rivers, and lakes [9,10]. Atmospheric N₂O increased globally by 1 ± 0.3 ppb yr$^{-1}$ from 2010 to 2018 [11]. This increment resulted from the increased use of inorganic fertilizers and manure [12–14], and those sources are all subject to significant uncertainty. The significant increasing trend in agricultural soil emissions from Southern Asia, which includes China and India, is mostly due to the rise in nitrogen fertilizers in these developing economies [13–15].

Recently, to better quantify these emissions, the importance of verifying the bottom–up inventories using top–down constraints provided by atmospheric N₂O measurements is increasing [15,16].

To reduce the uncertainties of those approaches, precise and reliable measurements of atmospheric $N_2O$ are necessary. Normally, $N_2O$ is analyzed by gas chromatography with an electron-capture detector (GC-ECD), with its advantages of low cost compared to other techniques. This method has been used widely by many researchers over the past three decades [17]. However, frequent calibrations for this technique are necessary to correct for nonlinearity and drift. The time taken to run samples (4–6 min per sample, at least) is a major disadvantage of this technique.

Optical techniques based on laser-absorption spectroscopy for gases were commercialized recently. One of those instruments, i.e., off-axis integrated cavity output spectroscopy (OA-ICOS), measures $N_2O$, CO, and $H_2O$ in the 4.6-μm wavelength region with a tunable laser and an optical cavity. This analyzer requires less maintenance at a station, with a measurement every 10 s. Previous studies have compared this new technique to conventional gas chromatography with an electron capture detector (GC-ECD); however, they were carried out inside laboratories [18] and not at field measurement sites. Since the instrumental performance depends on the local environment, it would be meaningful to compare the performance at an in-situ monitoring station. The Korea Peninsula is affected by not only local sources and sinks but, also, by outflows from the Asian continent, and hence, it shows large fluctuations with very distinct seasonal characteristics [19].

Quasi-continuous measurements of atmospheric $N_2O$ in South Korea by the Korea Meteorological Administration (KMA) started in 1999 on the West coast (Anmyeondo Station, 126.32° E, 36.53° N, 47 m a.s.l.), in the south (Jejudo Gosan Suwolbong Station, 126.16° E, 33.30° N, 71.47 m a.s.l.), and in the east of the peninsula (Ullengdo Station, 130.90° E, 37.48° N, 220.9 m a.s.l.) in 2012. Atmospheric $N_2O$ was measured by gas chromatography with microelectron capture detection (GC-μECD) at all three stations to understand its sources and sinks as covered by the Korean Peninsula. Under the World Meteorological Organization (WMO), the Global Atmosphere Watch Programme (GAW) was initiated in 1989, and there are around 81 in-situ stations to monitor $N_2O$ now [20]. Many stations still observe atmospheric $N_2O$ with GC-ECD rather than the spectroscopy method. For the confirmation of quality control/assurance, the WMO/GAW Central Calibration Laboratory (CCL) has hosted inter-comparison experiments for GAW stations/laboratories every 4–6 years since 1984. In 2015, 27 stations/labs attended the $N_2O$ comparison experiment, and only three were within the WMO compatibility goal of ±0.1 ppb. The major cause for the deviation was GC-ECD's nonlinearity and drift. Therefore, if we verify the performance of the OA-ICOS, it can help to measure atmospheric $N_2O$ with high accuracy.

In this paper, we describe experiments conducted to establish the linearity, repeatability, and reproducibility of both GC-μECD and OA-ICOS. Based on those tests, the uncertainty factors were estimated and a calibration strategy suggested for each instrument. Two instruments were installed in parallel at the Jeju Gosan Suwolbong (JGS) GAW Station (126.16° E, 33.30° N), and their performances were compared for almost two years. We also investigated atmospheric $N_2O$ characteristics with enhancement values above the temperate Northern Hemisphere (TNH) from 2018 to 2019.

## 2. Sampling Site and Measurement System

### 2.1. Sampling Site

JGS is located in the western part of Jeju Island, which is the biggest volcanic island (1845.88 km$^2$) situated in the southwest of South Korea, which is about 90 km from the mainland. Jeju is popular with tourists regardless of the season, while the region of Suwolbong is famous as a global geopark due to outcrops of volcanic deposits exposed along the coastal cliff where JGS is located. The side of the station from the southwest to the northwest is open to the sea, where there are volcanic basalt rocks. The sea to the south is connected to the East China Sea, and the sea to the west is linked to the Yellow Sea. Next to JGS is the widest plain in Jeju, where potatoes, garlic, and onions are harvested (Figure 1). Therefore, when the wind speed is less than 5 m/s under stagnant conditions, high $N_2O$ concentrations are observed at the station.

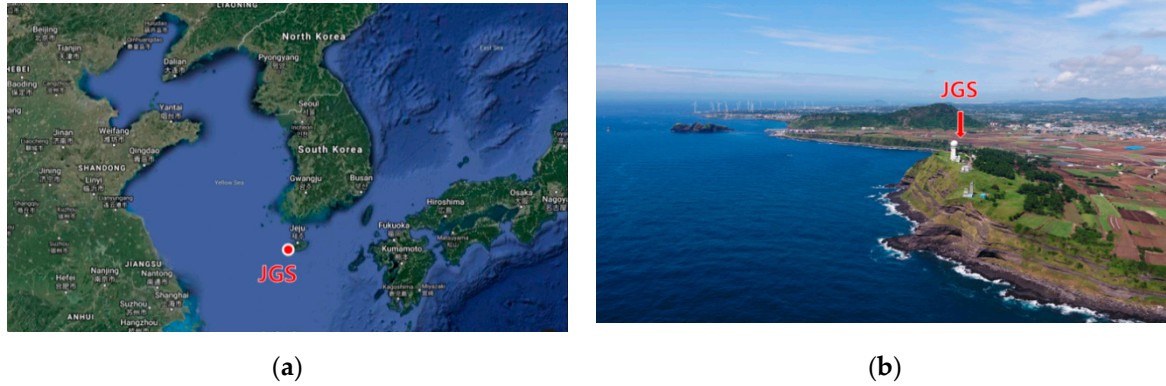

**Figure 1.** The location of the Jeju Gosan Suwolbong (JGS, 126.16° E, 33.30° N) Station on a map (**a**) and its environment around the station (**b**). The inlet height was 6 m until 2016, and it was changed to 12 m in 2017.

## 2.2. Measurement System

GC-µECD (Agilent 7890A, CA, USA) and OA-ICOS (Los Gatos EP-30, CA, USA) collect ambient air from an intake at 12 m height at JGS. Before the air is injected into the instruments, all the sampled air is dried cryogenically to −50 °C [19]. We do not apply the water vapor correction provided by the OA-ICOS manufacturer. Therefore, the water vapor effects on the $N_2O$ observations are negligible [21]. The analysis condition for GC-µECD and instrument specification for OA-ICOS are presented in Table 1.

**Table 1.** The analytical conditions for the microelectron capture detector method (GC-µECD) and the performance specification for off-axis integrated cavity output spectroscopy (OA-ICOS).

| GC-µECD (Agilent 7890A) | |
|---|---|
| Sample loop | 5 mL |
| Column | Main: Porapak-Q (Restek, PA, USA), 80/100 mesh, 12 ft, 2.0 mm Interior Diameter (I.D.), 1/8"Outside Diameter (O.D.)<br>Post: Porapak-Q (Restek, PA, USA), 80/100 mesh, 6 ft, 2.0 mm I.D., 1/8" O.D. |
| Oven Temp. | 60 °C |
| Detector Temp | 375 °C |
| Carrier gas flow rate (P5, $CH_4$ 5% in Ar) | Main: 45 psi<br>Post: 42 psi |
| Make-up flow rate | 5 mL/min |
| Sample flow rate | 100 mL/min |
| OA-ICOS (Los Gatos EP-30) | |
| Cell size | 434 mL |
| Cell pressure | 85 ± 0.1 Torr |
| Cell temperature | 45 ± 0.005 °C |
| Sample flow | 300 mL/min |

GC-µECD has been a widely used instrument for atmospheric $N_2O$ measurements. Two Porapak-Q columns, used as a precolumn and a main separation column, were plumbed with a 5-mL sample loop through two 6-port gas sample valves (Valco Instrument, TX, USA). Those valves allow automated sample injections, backflush of the precolumn, and the separation of $N_2O$ from other components in the air. Carrier gas is 5% $CH_4$ in Ar (P5). The main and post-columns for the backflush prevent other ECD-sensitive species such as CFCs (Chlorofluorocarbons) from reaching the detector. Therefore, $N_2O$ can

be measured without adjusting the oven temperature. This results in a shorter total run time than a conventional valve system [22,23]. The retention time of the $N_2O$ peak is normally ~around 13 min in this system, and the total run time is less than 20 min. Therefore, this method allows three measurements per hour. Due to drift, one of the three samples every hour is air from a reference cylinder (one-point calibration).

OA-ICOS is a recently commercialized instrument. This instrument, which is used in this study, is made by Los Gatos Research (LGR, CA, USA). For OA-ICOS, the laser light enters the high-finesse cavity at a non-zero angle, generating a high density of transverse cavity modes [24]. This off-axis arrangement allows a range of beam incidence conditions to be used, making the analyzers less susceptible to changes in the alignment of the mirrors and allowing sensors to be simpler to operate and more robust for operation in the field. Here, we used the LGR-EP30 with a 434 mL cell, operated at 85 ± 0.1 Torr and 45 ± 0.005 °C. We calibrated the instrument weekly with one standard. When we tested the time for the OA-ICOS output to stabilize after gas switching, the stabilization takes at least 250 s (around 4 min) from high $N_2O$ (351.09 ppb) to zero air. After 250 s, the deviation ranges between 0.01 to 0.09 ppb for high $N_2O$ and 0 to 0.02 ppb for zero, respectively. For the laboratory test, OA-ICOS takes from 2 to 3 min, which is similar to this study [18]. Therefore, we carried out the experiments for linearity, repeatability, and reproducibility or calibration for OA-ICOS after the sample cell was flushed enough for 10 min.

We used tertiary standards on the WMO-X2006A $N_2O$ scale certified by the National Oceanic and Atmospheric Administration (NOAA), as in the WMO/GAW/CCL. The scale was propagated with an uncertainty of ±0.09 ppb at the 68% confidence level [25].

All data were collected and stored at the KMA/National Institute of Meteorological Sciences (NIMS) in Jeju, South Korea. Our data selection was implemented in two steps, automated (AQC) and manual (MQC), according to the instrument. (1) AQC: Here, the measurements were automatically flagged for GC-μECD when $N_2O$ followed an outlier over the range 0–600 ppb, the difference in retention times of consecutive $N_2O$ measurements was more than 1 min, and the ratio of the consecutive sample peak area was less than 0.8. For OA-ICOS, auto-flagging occurred when $N_2O$ was an outlier, as with GC-μECD, and measured cell pressure and temperature were outside the acceptable ranges provided by the manufacturer. (2) MQC: For both instruments, the manual flags were assigned by technicians at JGS according to the logbook based on the inlet filter exchange, diaphragm pump error, low flow rate, dehumidification system error, and calibration periods, such as participation in comparison experiments [19]. After these flagging steps, the data were designated Level 1 (L1). Data with flags were reviewed by scientists at NIMS, and valid data were selected as Level 2 (L2). We used L2 hourly data for the continuous comparison experiment in Section 4.2.

## 3. Analysis Method of Measurement Uncertainty Factors

### 3.1. Linearity

The test is based on the premise that responses of the instrument change linearly with changes in the abundance of target species, or, to reiterate, the drift-corrected instrument response is the same within the range of the $N_2O$ of interest (313.85 to 351.09 ppb).

The GC-μECD peak areas of four standards were normalized due to the instrument drift based on injections from a reference gas between aliquots of standard gas. Therefore, an A-B-A' sequence was used for this test, where A was a tertiary standard and B was a reference gas to correct the drift. However, the OA-ICOS did not need the drift correction, since the drift in an hour was not significant (see Section 4.2).

The uncertainty of linearity is defined as the residuals from the regression curve and calculated by the equation below:

$$U_r = \sqrt{\frac{\sum R_i}{N-2}} \tag{1}$$

Here, $N$ is the number of cylinders (here, 4 cylinders), and $R_i$ the residuals of each cylinder. $R_i$ is the difference of the laboratory standard minus the value from the regression curve in Figure 2.

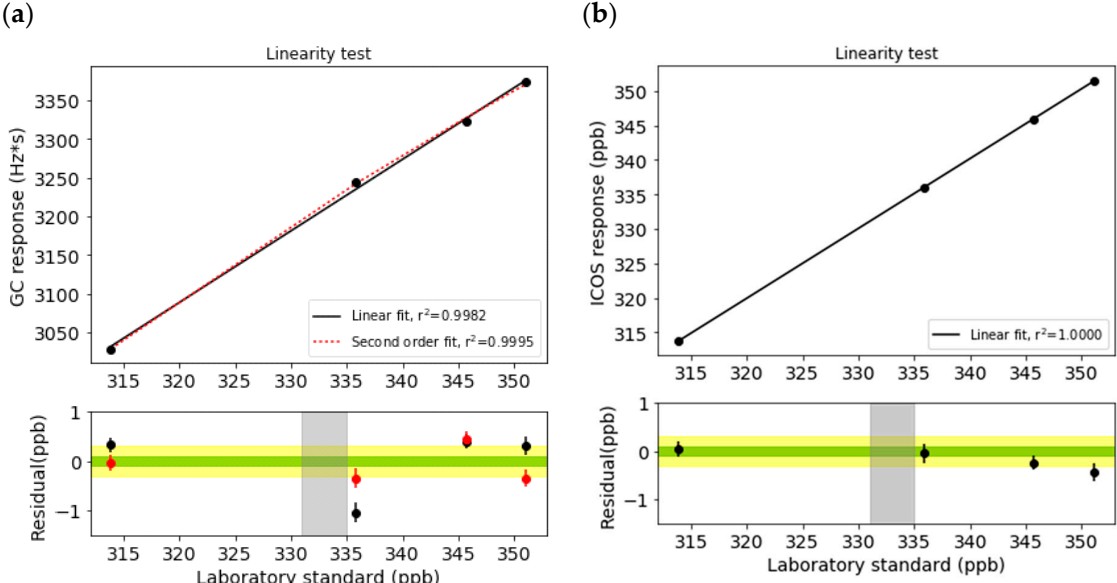

**Figure 2.** Linearity test of (**a**) the microelectron capture detector method (GC-µECD) and (**b**) off-axis integrated cavity output spectroscopy (OA-ICOS) for $N_2O$. The $x$-axis is the assigned values of the laboratory standards, and the $y$-axis is the drift-corrected instrument response (top). Residuals are differences, the laboratory standard minus the value from the regression (bottom). The black line and dots represent the linear regression function and residuals, respectively, while red represents a second-order polynomial and its residuals. For (**a**) GC-µECD, $m$ is 9.2350 ($r^2 = 0.9982$), while it is 1.00116 for OA-ICOS ($r^2 = 1$) when the linear regression function is applied. The red lines and dots are derived from a second-order polynomial. The World Meteorological Organization/Global Atmosphere Watch Programme (WMO/GAW) compatibility goal (±0.1 ppb) and extended compatibility goal (±0.3 ppb) are shaded. The grey band represents the observed $N_2O$ level at JGS.

### 3.2. Repeatability

Repeatability ($U_p$) is determined from the standard deviation ($1\sigma$) of the same standard gases in an hour and expressed by a pooled standard deviation:

$$U_\text{p} = \sqrt{\frac{\sum_{i=1}^{N} Ni \times Si^2}{4Ni - Nt}} \tag{2}$$

where $S_i$ is the standard deviation of 1 h averages of the same laboratory standard measurements, $N_i$ the number of data over 1 h (based on 10 s intervals for OA-ICOS and 3 peaks for GC-µECD), and $N_t$ is the total number of cylinders: 4. While testing the linearity for two instruments, we implemented the repeatability test simultaneously.

### 3.3. Reproducibility

Long-term reproducibility ($U_L$) was defined as the drift with the surveillance cylinder during 1 week with the same $N_2O$-level standard gas and tracing the values for two months for both GC-µECD and OA-ICOS. Since positive and negative values were found for $U_L$, we used the following equation:

$$U_\text{L} = \sqrt{\frac{\sum_{i=1}^{N} (x_i)^2}{N}} \tag{3}$$

where $x$ is the drift between weekly injections, and $N$ is the total number of data (weekly data for two months).

## 4. Results and Discussion

### 4.1. Measurement Uncertainties

GC-µECD was nonlinear in the target range, meaning that one-point calibration was not sufficient, while OA-ICOS showed a linear response against the reference gases (Figure 2). Even if a second-order polynomial was applied as the regression curve for GC-µECD, the residual (differences between the theoretical value from the function and observed values from the instrument) was still more than the WMO/GAW compatibility goal (the horizontal shaded area in Figure 2) in a certain range.

Such differences were observed in past inter-comparison experiments through audits implemented by the Swiss Federal Laboratories for Materials Science and Technology (Empa) [26] and the Round Robin Test hosted by WMO/GAW/CCL with travelling cylinders, which are traceable to the WMO scale [25]. In those inter-comparison experiments, one-point calibration was used, which cannot cover the target range, and hence, it showed a significant difference in the experiments. The nonlinear response of the GC-µECD is well-known [23,25], and this could bias atmospheric measurements. Therefore, nonlinearity should be assessed when developing a calibration strategy to monitor atmospheric $N_2O$ using GC-µECD.

For OA-ICOS, we sampled one standard and used the calibration function provided by the manufacturer for the linearity test. We used 335 ppb for this calibration, and this covered the range of 310 ppb to 345 ppb. However, this can be out of range of the WMO/GAW compatibility when $N_2O$ is more than 345.7 ppb, as indicated by the increasing residuals as the $N_2O$ increased. This suggests that the monitoring stations in the regions that observe a wide range of atmospheric $N_2O$ should conform to the calibration function provided by the manufacturer with a one-point standard that covers the anticipated range of atmospheric $N_2O$.

The $U_r$ from OA-ICOS is 0.35 ppb. For GC-µECD, the $U_r$ from the linear function is 0.85 ppb; however, we confirmed that it can decrease to 0.48 ppb when the second-order polynomial regression curve is used to describe the detector response to $N_2O$ (Table 2). To decrease the $U_r$, bracketing standards are recommended, at least when using GC-µECD.

**Table 2.** Estimated uncertainty factors in the measurements of $N_2O$ from each instrument. Units are ppb. All terms have a 68% confidence interval. The values in brackets are derived from the GC-µECD applied by the second-order polynomial regression curve. $U_r$: uncertainty of linearity, $U_p$: repeatability, and $U_L$: long-term reproducibility.

| Uncertainty Factors | Analyzer | | |
|---|---|---|---|
| | **OA-ICOS** | **GC-µECD** | **Relevant Equation** |
| $U_r$ (second-order polynomial) | 0.35 | 0.85 (0.48) | Section 3.1 Equation (1) |
| $U_p$ | 0.03 | 0.26 | Section 3.2 Equation (2) |
| $U_L$ | 0.46 | 1.19 | Section 3.3 Equation (3) |

For the gap between the minimum and maximum standard deviations over an hour, the value from the GC-µECD (0.20 ppb to 0.36 ppb) was almost five times greater than OA-ICOS (0.02 ppb to 0.04 ppb), while $U_p$ was ±0.26 ppb and ±0.03 ppb for the GC-µECD and OA-ICOS, respectively (Table 2). The previous study reported that $U_p$ derived from the GC-µECD was less than 0.4 ppb with a 20 min injection of standard gases [27] and 0.1 ppb with a 1 h average [18]. The short-term continuous measurement repeatability incorporating the GC-µECD was 0.1 ppb to 0.3 ppb [28–30]. The $U_p$ from OA-ICOS was similar to that of the previous study at ±0.05 ppb with a model EP-40 in the lab test [18].

The $U_L$ from OA-ICOS and the GC-µECD were ±0.46 ppb and ±1.19 ppb, respectively (Table 2). The long-term drifts for 10 days from OA-ICOS with models EP38 and EP40 were reported as ±0.76 ppb

and ±0.31 ppb, respectively, inside the laboratory [18]. Considering that our model is different from the reference and long-term drift is defined as one week, the $U_L$ in this study is at a similar level to that of previous studies. On the other hand, the $U_L$ is behind the WMO/GAW compatibility goal for $N_2O$, suggesting that the calibration frequency should be adjusted. When we track the drift everyday with the same cylinder over a week, the drift increases to ±0.23 ppb after three days. Note that a previous study suggested twice a day [18], such that the estimation of the $U_L$ is necessary for deciding the calibration frequency of OA-ICOS. For the GC-μECD, the other study showed that the drift during two–four days was 0.2 ppb to 0.3 ppb [28], which are smaller than the value obtained from this study. It was suggested that the $U_p$ and $U_L$ from the GC-μECD can be decreased to 0.05 ppb (1σ) and 0.1 ppb (1σ), respectively, under ideal conditions with no other factor influencing those values other than the electronic noise [25]. However, the GC-μECD in a field cannot be controlled only by electronic noise. At the JGS, 23.3 ± 0.92 and 1008.5 ± 7.7 hPa in a year, the GC-μECD can be affected by these laboratory conditions. The variation of the GC-μECD sensitivity is related to variations in the temperature and pressure in the laboratory, indicating that 2.15% of sensitivity variations, with ±1.5 of temperature and ±10 hPa of pressure, respectively [31].

### 4.2. Inter-Comparison Experiment between OA-ICOS and the GC-μECD

As explained in Section 2.2, we applied one-point calibration for both instruments. The frequency of the reference gas injection to correct the drift was 1 h for the GC-μECD and a week for OA-ICOS.

We implemented parallel measurements on the two instruments for 22 months, with measurements every 10 sec for OA-ICOS and two peaks for the GC-μECD, respectively, at the JGS and the compared the hourly data (see Section 2.2). $N_2O$ from the two instruments showed R = 0.8. The value from the GC-μECD was lower than that from OA-ICOS during the experimental period, as indicated by the mean difference (OA-ICOS minus the GC-μECD) of 0.17 ± 0.9 ppb (Figure 3). This mean value was in the WMO/GAW extended the compatibility goal of ±0.3 ppb, though the standard deviation was quite large.

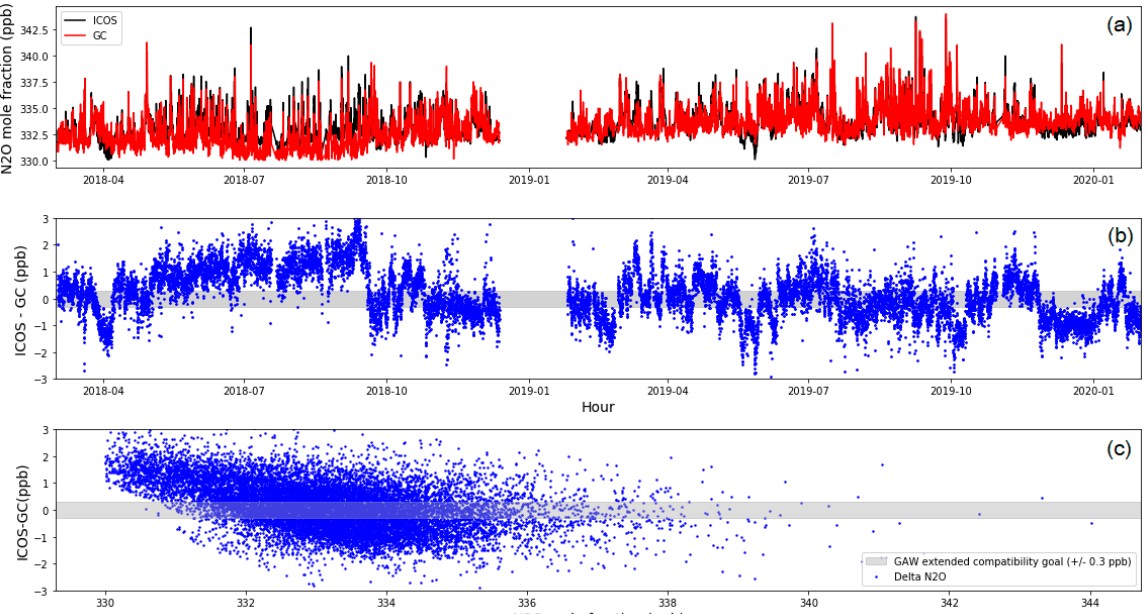

**Figure 3.** Comparison of $N_2O$ measured by OA-ICOS and the GC-μECD during 1 March 2018 to 1 January 2020. (**a**) Time series of OA-ICOS (black) and the GC-μECD (red) hourly mean, (**b**) time series of $\Delta N_2O$ (OA-ICOS minus the GC-μECD) over the same period, and (**c**) the $\Delta N_2O$ scatterplot according to the atmospheric $N_2O$ level measured by the GC-μECD. Note from 23 December 2018 to 25 January 2019, the OA-ICOS pump had a malfunction, and so, data were not compared during that period.

The differences might be related to the calibration methods for each instrument. As we described above ($U_L$), the calibration frequency should be shorter than a week for OA-ICOS. The difference between the two instruments increased (up to around 3 ppb) for $N_2O$ by less than around 332 ppb, mainly. This might be derived from the nonlinear characteristic of the GC-μECD, while OA-ICOS performed well over the same range, as seen in Figure 2. Since the level of standard gas was 334 ppb for the one-point calibration of the GC-μECD, as described in Section 3.2, a nonlinearity of less than 332 ppb might not be corrected. Therefore, it was confirmed that the calibration strategy is important for the precise measurements. For the GC-μECD, it also could be affected by laboratory conditions such as the variations of temperature and pressure that cause the differences from OA-ICOS.

When we compared the standard deviation of an hour with the observed ambient $N_2O$ during the period, the median value of OA-ICOS was 0.09 ppb, while for the GC-μECD, it was 0.25 ppb. Even though the same ambient samples were sampled by both instruments, the standard deviation from OA-ICOS was lower than that from the GC-μECD. This is reflected in the $U_p$ factor from the instrument and the number of samples under the same condition, indicating that the precision of OA-ICOS was better.

### 4.3. Enhancement $N_2O$ Values

The enhancements of $N_2O$ (ex$N_2O$) observed by the GC-μECD and OA-ICOS were compared to see the local characteristics in each season. The ex$N_2O$ was expressed as the difference from the zonal mean over 17.5° N to 53.1° N (temperate Northern Hemisphere, TNH) observed by the NOAA (Figure 4). This is to see local and seasonal variations of the JGS $N_2O$ as removing, representing the zonal growth rate and seasonal characteristics from 17.5° N to 53.1° N equally from the values measured by the two instruments. The amplitude of the TNH is ~0.5 ppb, with the lowest in August and highest in March. Even when this amplitude is considered, the JGS is almost always higher than the TNH.

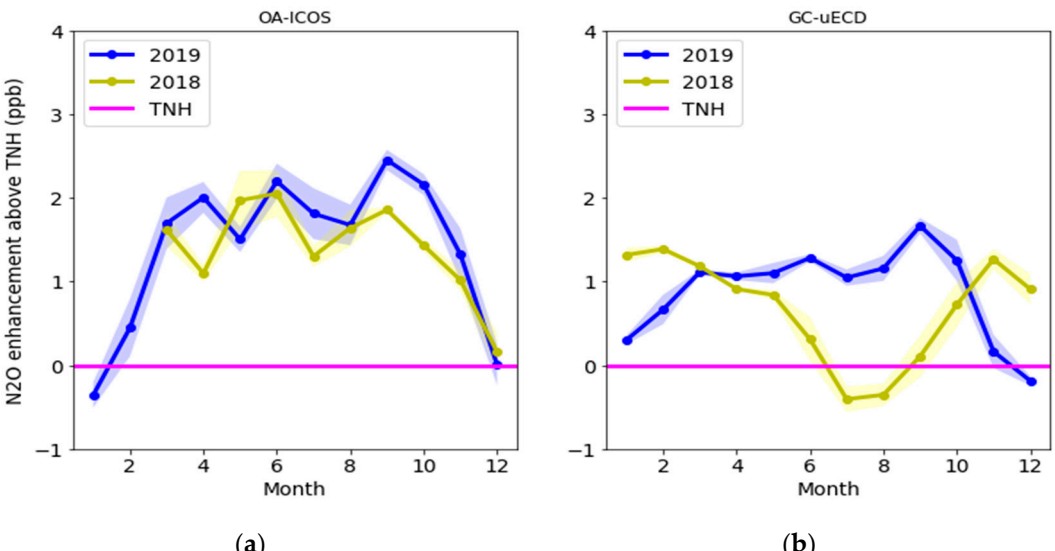

**Figure 4.** The JGS $N_2O$ enhancement values (ex$N_2O$) observed by OA-ICOS (**a**) and the GC-μECD (**b**) in 2018 (yellow line) and 2019 (blue line), respectively. The yellow line represents the atmospheric $N_2O$ observed by the GC-μECD and blue line by OA-ICOS. The magenta line is the zonal mean atmospheric $N_2O$ for 17.5° N to 53.1° N obtained from the National Oceanic and Atmospheric Administration (NOAA). The uncertainty bands for the GC-μECD and OA-ICOS were standard deviations of the monthly mean. TNH: temperate Northern Hemisphere.

The JGS monthly mean was calculated as follows: (1) to select for the background conditions at the JGS, we considered the differences between the consecutive hourly averages of the L2 data, which was

described in Section 2.2 as less than 0.5 ppb (1$\sigma$). (2) The selected hourly data were used to calculate the monthly means with the method by [32] to represent the regional baseline, thereby reducing the noise due to synoptic-scale atmospheric variability and measurement gaps.

The exN$_2$O from OA-ICOS in the winter (December to February) is similar or lower than the monthly mean of the TNH through 2018 to 2019, even though the data for two months, from January to February of 2018, were missing (Figure 4, left). This might result from seasonal wind characteristics and local agricultural activities. The JGS could collect well-mixed air in the winter rather than in the other seasons due to a strong Siberian high from the continent. As described in Section 2, there is a large plain around the JGS and Jeju that raises two crops a year due to its warm weather. Therefore, high N$_2$O episodes were observed, regardless of the season. Those environments allow atmospheric N$_2$O to be greater than the TNH in the spring to autumn, while a similar level of N$_2$O to the TNH was observed in the winter.

However, the characteristics of exN$_2$O from the GC-$\mu$ECD are different each year, as shown by the N$_2$O decrease in the summer of 2018 (even lower than the TNH) and winter of 2019. This contrasts the pattern from OA-ICOS.

On the other hand, the difference in the annual mean was 0.6 ppb between the two instruments (OA-ICOS minus the GC-$\mu$ECD). The differences from the monthly mean were observed continuously from 2018 to 2019. It increased when the monthly N$_2$O level decreased, and, especially, significant differences occurred for N$_2$O of less than 322 ppb (June to August 2018), which was quite similar to the parallel comparison, indicating the bias reflected the monthly variation. For example, the GC-$\mu$ECD N$_2$O was similar to the TNH N$_2$O in the summer of 2018, and this level might be underestimated compared to the values from OA-ICOS due to the nonlinear characteristic of the GC-$\mu$ECD.

Therefore, the measurement uncertainty described in Section 4.1 finally resulted in the over/underestimations of large-scale characteristics of atmospheric N$_2$O.

## 5. Conclusions

To understand the sources and sinks of atmospheric N$_2$O and to reduce the uncertainty in the inverse assessments based on atmospheric observations, precise measurements are important. In this study, we investigated the conventional GC-$\mu$ECD and the advanced OA-ICOS analytical methods for atmospheric N$_2$O measurements.

Overall, the linearity, repeatability, and reproducibility tests showed that the OA-ICOS has a more robust performance than that of the GC-$\mu$ECD in the in-situ station. Since these factors affect the measurement uncertainty, the calibration strategy should be well-established to reduce it.

For the OA-ICOS, one-point calibration is sufficient to monitor the atmospheric N$_2$O. However, the calibration should be implemented on a less-than-weekly basis. When the drift was tested by injecting the same cylinder every day, the value showed that the drift increased to $\pm$0.23 ppb after three days (WMO/GAW compatibility goal, $\pm$0.1 ppb and extended goal, $\pm$0.3 ppb). This means that the calibration frequency should be shorter than three days. Note, however, the environment is different according to the in-situ stations, so that the linearity, repeatability, and reproducibility tests should be performed at each station prior to the routine measurements for high-quality measurements.

It is suggested that the GC-$\mu$ECD needs at least two reference cylinders to set the target levels within an hourly calibration. This is to correct its nonlinear characteristics and large drifts, even within an hour. In this study, we used one reference cylinder. Since three peaks are observed in an hour, if we apply the two-point calibration method, we can get only one atmospheric data among the three. According to the uncertainty factors, the value derived from reproducibility was greater than the other factors not only for the GC-$\mu$ECD but, also, for OA-ICOS, indicating that the drift correction is the most important. Therefore, we put the priority on the drift correction. This calibration strategy resulted in biases in the nonlinear range, as described in Sections 4.2 and 4.3. Therefore, for the GC-$\mu$ECD, both drift and nonlinear range corrections are necessary to get more accurate data.

For both instruments, the uncertainty value was high in the order of the $U_L > U_r > U_p$. This means that, when we adjust the calibration frequency, we can reduce the total measurement uncertainty. For the GC-μECD, however, even after correcting the short/long-term drift with an hourly reference gas injection, the uncertainty from linearity resulted in the bias of the quasi-continuous measurements for atmospheric $N_2O$.

Hourly L2 $N_2O$ data were compared between two instruments for 22 months, and the difference was $0.17 \pm 0.9$ ppb, with a correlation R of 0.8. When the monthly ex$N_2O$ were compared, OA-ICOS showed a constant seasonal pattern, while the GC-μECD varied year by year. Even though the two instruments observed the same atmospheric $N_2O$, the hourly data and monthly background levels were quite different, especially in the range related to the nonlinear characteristics of the GC-μECD. Finally, we confirmed that the seasonal and regional background levels can be underestimated or overestimated because of the instrument performance.

Through this paper, the OA-ICOS showed better performance in monitoring atmospheric $N_2O$, with lower measurement uncertainty in the in-situ station, especially in regions with large fluctuations that reflect not only synoptic conditions but, also, local activities. Additionally, the appropriate calibration method for each instrument is strongly recommended for the high-quality measurements data.

**Author Contributions:** Conceptualization and methodology, H.L.; writing and validation, H.L.; investigation, M.K. and S.K.; data handling, W.S.; reviewing, H.L., M.K., S.K.; editing H.L.; supervision, Y.-S.P. All authors have read and agreed to the published version of the manuscript.

**Funding:** This research was funded by the Korea Meteorological Administration Research and Development Program "Research and Development for KMA Weather, Climate, and Earth system Services—Development of Monitoring and Analysis Techniques for Atmospheric Composition in Korea" under grant 1365003239 (KMA2018-00522).

**Acknowledgments:** We thank the NOAA Global Monitoring Laboratory Carbon Cycle Greenhous Gas group for providing the temperate NH zonal average of $N_2O$.

**Conflicts of Interest:** The authors declare no conflict of interest.

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
