# Peer review of "Comparison of GC-μECD and OA-ICOS Methods for High-Precision Measurements of Atmospheric Nitrous Oxide (N2O) at a Korean GAW Station"

_atmosphere, doi:10.3390/atmos11090948_

Round 1

Reviewer 1 Report

The authors have addressed my comments

Author Response

Thank you so much for your review and accepting this manuscript.

Haeyoung Lee, on behalf of co-authors

Reviewer 2 Report

General Comments:

The paper is well written and the overall scientific presentation and comparison between the two instruments is sound and well done.  

The authors need to carefully edit the paper, as some of the phrasing is awkward and does not quite make sense. However, these changes are minor. 

Overall, this is a nice, succinct paper, and I recommend it be published after very minor corrections are made.  

Specific Comments:

Line 78- 81: This statement is confusing.  Precision represents the ability of an instrument to get the same result for a given unknown measured, while accuracy describes the ability of an instrument to measure the true value.  Laboratories can easily have precise values that are inaccurate.  Please clarify whether the ECD instruments actually have poor precision (large variability in measurements internal to a laboratory, or if they just don’t agree with each other.  

Line 82: Possibly use precision and accuracy instead of reproducibility and repeatability.

Line 82 - 88: Consider rephrasing.  What are uncertainty factors vs the values you reference?

Line 85-86: You already defined GAW.  No need to redefine here.

Section 2.1:  While I enjoyed the wonderful description of Jeju Island, much of the description does not seem to pertain to the interpretation of the data.  It is useful to know that there is farmland nearby given the N2O measurements, but it would be more helpful to the reader to know more about the features of the site that affect the measurements.  

Line 127-128: revise these sentences.  Repetitive/redundant language.

Line 155: reference appropriate section that describes the continuous experiment.  This is helpful for the reader.

Figure 3: Suggest that the authors convert the GC response axis in the top panel of plot “a” to ppb, since the other 3 panels are in ppb. 

Section 3: Please add all values of “N”.  There are some references to the number of tanks used, but not the number of data.  It is difficult to assess the statistics you present without knowing the total number of data involved in each test.  For example, how many Non-linearity runs were done?

Line 263: This statement is confusing.  You state in Section 4.1 several times that a one point calibration is insufficient for the ECD.  You cite section 2.2, but in this section, you simply state that a one point calibration is used, but not why a 2 point calibration is not implemented.  Please clarify this.

Author Response

Thank you so much and please see the attachement.

This manuscript is a resubmission of an earlier submission. The following is a list of the peer review reports and author responses from that submission.

Round 1

Reviewer 1 Report

Regarding the manuscript Sensors-829500, it is an interesting technical approach to the optimization of nitrogen protoxide monitoring, an impacting GHG.

While the manuscript has scientific merit, the main issue is related to the structure, which is quite confusing for the reader because results are mixed with methodology, and discussion is poorly presented. Furthermore, the writing style could be improved. I recommend the support of a native English speaker before re-submitting the revised manuscript.

I have provided some details that could improve the current version towards a publishable form (a non-exhaustive list).

Title: I suggest “Comparison of GC-μECD and OA-ICOS methods for high-precision measurements of atmospheric nitrous oxide (N2O) at a Korean GAW station”

Abstract

L14 Avoid excessive use of “Here…”, check throughout the manuscript; and at this line use comma Here, …

L16 …Jeju Gosan Suwolbong station

L17 reformulate (e.g. Linearity, repeatability, and reproducibility derived from the two instruments were  compared.)

L30 Use point at the end of the sentence.

Keywords: Add more relevant keywords e.g. GHGs, climate change (only 3 may result in lower accessibility on search engines)

Introduction section could be extended by providing more insights on the monitoring of GHGs in Korea (history, present, and perspectives). In my opinion, this will increase the value of the work. Moreover, GAW program should be described Example from other papers:

“In the late 1960s, the Background Air Pollution Monitoring Network (BAPMoN) was established. It has focused on precipitation chemistry, aerosol and carbon dioxide measurements, and included regional and background stations. In 1989, two observing networks namely BAPMoN and GO3OS (Global Ozone Observing System) were consolidated into the current World Meteorological Organization (WMO) Global Atmosphere Watch (GAW) program.” Source: http://eemj.eu/index.php/EEMJ/article/view/2784

L35 Why N2O is a powerful gas? Replace powerful with other appropriate characteristics (i.e. powerful oxidizer)

L44 This increment…

L54 references are required for this statement

L58 One of those instruments i.e., off-axis integrated …

L78 In this paper, we…

L79 Delete “the”

Methods

L94 switch images in figure 1 (left – the map; right – the picture of JGS)

L97 Consider a reformulation of “2.2. The measurement system, instrument description, and data QA/QC method”. Maybe “Instrumentation and methods of analysis” would fit better.

L103 are presented (not suggested!!)

L104 use bold also for OA-ICOS

L106 replace well-known with widely used

L108 comma before which

L110 use point after (P5)

L106-115 the paragraph needs rewriting using shorter sentences and past tense.

Methods section misses the description of statistical analyses. I recommend presenting more statistical indicators and residuals of analyzed time series using tables in the results.

Results

Since the manuscript does not have a discussion section and the current results comprise discussions as well, I recommend renaming this section: Results and Discussion. The section contains paragraphs that describe methods, which need to be moved to methods. In my opinion, the whole section should be rewritten.

L143-145 are not results

For this paragraph, commenting on figure 1 would be more suitable and present what is important?

L174-178 should be moved to methods in the statistical analysis description, which is missing now.

L181-199 are not results. All equation should be mentioned in methods

Hint for authors: to optimally address the presentation of results, comment on the tables and figures extracting the key values and findings. Then, compare to other works from literature for a proper discussion. The section should be completed with limitations and future work.

L316   4. Summary and conclusion should be Conclusions

Part of the text would be more suitable for discussion. Conclusions should be key findings and recommendations regarding the practical use of the OCA-ICOS technique.

Reviewer 2 Report

Review of Comparison of GC-ECD and OA-ICOS for high-precision measurements of atmospheric nitrous oxide (N2O) at a Korea GAW station

This study compared the precision and reproducibility of N2O concentration measurements using a conventional gas chromatography (GC) method and a laser-absorption based analyzer. The authors find that the laser analyzer has a lower uncertainty compared to the GC, determined as a function of the residual’s arounds known standards. Although the authors also report a variety of other metrics of performance, the uncertainty is used as the main basis for preferring the laser analyzer over the GC.

Overall the figures are fairly well presented, the methodology is mostly sound (see questions below), and the conclusions are supported by the data. However, I am not convinced that the paper offers much new insight beyond previous analyzer comparisons (e.g., Lebegue et al. 2016 - doi:10.5194/amt-9-1221-2016). This seems like a very useful report for internal laboratory use, but more work is needed to make it broadly useful and impactful for an international journal. I had quite a difficult time reading the text, so I would encourage a careful proof-read focused on clarity of writing, before re-submission.

Major Comments/Questions

The introduction mentions the poor performance of the N2O stations/analyses in S Korea during a WMO round robin. Can you please return to this topic in the results or conclusions to give more context about how these results affect that outcome, or do your results provide any insight into why the results were poor before?

The biggest disconnect in the writing of this manuscript is the combination of the long-term monitoring data with the specific tests. Better structure may help the reader follow whether the long-term data prompted the testing, or whether the testing came first.

I am concerned about the use of the second-order polynomial in the linearity/uncertainty tests. Is this a typical approach? Does it not simply over-fit this particular curve? How do you know that the one point that moves closer to the polynomial fit wasn’t a reflection of “real” higher uncertainty?

In the results/conclusions can you comment on what are the sample size and inlet implications of the laser-based cavity system? How do these vary depending on a whole-air application versus other laboratory applications with smaller or discrete samples?

Can you provide a few clear recommendations or things to consider for both instruments, based on your results? For example, if one had the GC setup, how frequently should one run standard curves?

Please provide numbered references in the references section. I wasn’t able to check the papers your cited properly.

Minor Comments

Line 35: Please clarify if N2O radiative forcing is enhanced (ie post-Industrial) or total?

Figure 5: Why was the WMO round robin result so poor, if the long term measurements seem OK?

Figure 6: Are the plots based on wind velocity at the time of sample collection?
